# Use of simple analytical solutions in the calibration of Shallow Water Equations debris flow models

Bonomelli Riccardo[1], Marco Pilotti[1], Gabriele Farina[1]

[1]DICATAM, Università degli Studi di Brescia, Brescia, 25123, Italy

*Correspondence to*: Riccardo Bonomelli (riccardo.bonomelli@unibs.it)

**Abstract.** Modelling debris flow propagation requires numerical models able to describe the main characteristics of the flow, like velocity or inundation extent. Due to the complex physics involved, every numerical model is dependent from a set of parameters whose influence on the results is often not evident. In this contribution we propose simple analytical solutions based on the monophasic Shallow Water Equations for some of the most used rheological models (O'Brien & Julien, Voellmy,

Bingham and Bagnold) implemented in monophasic (FLO-2D, RAMMS, HEC-RAS, TELEMAC-2D) and biphasic commercial software (TRENT2D). These simplified solutions and their asymptotic uniform-flow like relationship are useful on one hand to speed up the calibration process, limiting the need to perform multiple simulations with unrealistic set of parameters and, on the other hand, as a benchmark for existing numerical methods. To further guide the calibration, a Sobol's sensitivity analysis has been performed to highlight which parameters of the considered equations influence the flow velocity

the most. Finally, as an example of application, the proposed methodology is validated on a real debris flow event occurred in Italy.

## 1 Introduction

The computation of hydraulic hazard related to debris flow is of paramount importance for risk mapping in mountain areas. In spite of their limitations with respect to models which better respect the physics of the process like the two-phase models (e.g.

Pitman and Le, 2005; Armanini et al., 2009; Pudasaini and Fischer, 2020) and multi-phase models (Pudasaini and Mergili, 2019), monophasic ones, based on Shallow Water Equations, are still widely used in practice and can be effective (e.g. Rickenmann et al., 2006) when suitably calibrated (e.g. Stancanelli and Foti, 2015). However, apart from theoretical limitations, when the adopted rheological relation is multiparametric, its calibration is not straightforward. In the following, dealing only with depth-averaged models, we will use the term "rheology" as a synonym for friction law. In the widely used

FLO-2D (O'Brien & Julien, 1988) model and also inside HEC-RAS Mud and Debris Flow (US Army Corps of Engineers, 2023), an empirical quadratic rheological relation is used, as a function of several parameters and of the sediment concentration. Simpler relations are the one proposed by Voellmy (Voellmy, 1955), implemented in the RAMMS model (Christen et al., 2010) and in HEC-RAS or the Bingham-like rheology (Malet et al., 2005; Begueria et al., 2009; Sauthier et al., 2015). The calibration of the rheological parameters is typically done, for past events, either by using experimental measurements or, more

commonly, by reproducing the extension of mapped inundated areas, the distribution of the debris flow volume and the timing

of the propagation, all of which information may not be available. However, over the last 10 years debris-flow monitoring techniques have greatly improved (e.g. Hürlimann et al., 2019) and many monitoring systems have been installed worldwide that can provide observations on debris flow velocity and the corresponding stage along a channel reach. The same information can also be provided by occasional recording and we believe that it can be extremely valuable in constraining the calibration

process. In this paper we propose to use a simple analytical solution for the uniform motion of a monophasic debris-flow with a general quadratic equation rheology along a channel of constant slope, to obtain the following results: 1) simplify the calibration by providing a priori insights on the role of the different rheological parameters on the modelled debris flow velocity, 2) identify a characteristic time of the model, to be used to evaluate the distance needed to reach the theoretical normal flow condition, and, in more general terms, to adapt the flow to bed-slope changes, 3) identify possible parameter combinations

that lead to flow slow-down, 4) investigate the different parameters sensitivity on the process. We believe that these observations can be used to simplify the calibration process when field data are scarce. As an example, we applied the described methodology to a real debris flow event occurred in Ono San Pietro, Italy, where footage of the debris flow was available.

## 2 Governing equations

The 1D Shallow Water Equations

$$
\begin{aligned}
h_t + (hu)_x &= 0 \\
(hu)_t + \left( hu^2 + \frac{1}{2} gh^2 \right)_x &= gh(S_b - S_f)
\end{aligned}
\tag{1}
$$

where $h \ [m]$ is the fluid depth, $u \ [m \ s^{-1}]$ is the depth-averaged flow velocity along the $x$ direction, $g \ [m \ s^{-2}]$ is the acceleration of gravity, $S_b \ [-]$ is the bed slope and $S_f \ [-]$ is the friction slope, can be used to model the propagation of a monophasic non-newtonian debris flow on a rigid bed if a suitable model is selected for the friction slope. Although a steady state condition is rare in natural debris flows, some field and laboratory measurements of debris flow velocity in a confined channel (e.g. Takahashi, 1991; Hungr, 2000; Lanzoni et al., 2017) have shown that, where the flow is fully developed (Bernard

et al., 2023) velocity records can show limited variations in terms of depth and velocity over distance. This constant velocity can be reproduced by a simple physically-based relationship of the asymptotic velocity of a debris layer with constant depth $h$ (measured normally with respect to the bottom), destabilized along an incline of constant slope. To account for cross-sections of limited width in a channel, it is possible to replace $h$ with the hydraulic radius (Perla et al., 1980). This analogy also provides a characteristic time for fluid acceleration. Using a Voellmy rheology, this approach was originally proposed for snow

avalanches (Voellmy, 1955; Perla et al., 1980; Pudasaini and Hutter, 2007) and, more recently, for debris flow (Christen et al., 2010; Kelfoun et al., 2011; Hergarten and Robl, 2015). The transient velocity $u$ of the debris layer can be obtained by simplifying system (1) for a layer of infinite length (or alternatively, by adopting a Lagrangian framework) as

$$
u'(t) = g(S_b - S_f)
\tag{2}
$$

Equation (2) can be solved analytically if the equation of the friction slope $S_f$ make it possible. A rather general rheology model is the quadratic one, according to which $S_f$ can be written as a function of velocity as

$$S_f = Pu^2 + Qu + R \tag{3}$$

where $P, Q$ and $R$ are functions of the flow depth or fluid properties. Combining Eq. (2) and (3) leads to a Riccati equation

$$u'(t) = A\,u^2(t) + B\,u(t) + C$$
$$A = -g\,P; \quad B = -g\,Q; \quad C = g\sin\vartheta - gR \tag{4}$$

in which $\vartheta$ [°] is the local inclination. Pudasaini and Krautblatter (2022) proposed a series of analytical solutions to estimate landslide velocity as a function of rather general rheologies where the linear term in the velocity is not included ($B = 0$) because this term is negligible compared to the quadratic one when extremely fast flows are considered (Perla et al., 1980). Models in which $B \neq 0$ were considered by Salm (1966) and Nishimura and Maeno (1989) but only for snow avalanches. However, the linear term $B$ may be important considering debris flows. Eq. (4) has some interesting properties. For instance, it is clear that $C$ must be $\neq 0$ if the motion starts from rest, as assumed in the following. The asymptotical velocity $u_\infty$ is provided by solving the quadratic equation obtained by setting $u'(t) = 0$ in Eq. (4). Accordingly, the discriminant $B^2 - 4AC$ must be positive for a uniform velocity to exist, which is always the case provided that all terms are different than zero and $C > 0$, which happens when the slope is steep enough for the motion to occur. Degenerate cases, i.e. when either $A = 0$ or $B = 0$, will be discussed separately. When $A \neq 0$, the solution $u(t)$ of Eq. (4) can then be obtained analytically with a standard integration of Riccati's equation, leading to

$$u(t) = -\frac{B}{2A} + \sqrt{|\varphi|}\,\tanh\left[\frac{(t+\varepsilon)}{T}\right]$$
$$\varepsilon = -\frac{1}{A\sqrt{|\varphi|}}\text{atanh}\left(\frac{B}{2A\sqrt{|\varphi|}}\right)$$
$$T = \frac{2}{\sqrt{B^2 - 4AC}}; \quad \varphi = \frac{4AC - B^2}{4A^2} \tag{5}$$

where $\varepsilon$ is a constant of integration computed using the condition that the initial velocity is zero, $u(0) = 0$, $T$ is the characteristic time of the flow, which measures how quickly the velocity reaches its asymptotic value and $\varphi$ is a short-hand for a more compact notation. The solution for the Voellmy model, investigated by Herganten and Robl (2015) as well as Pudasaini and Krautblatter (2022), implemented in the widely used RAMMS software (Christen et al., 2010), can be obtained using Eq. (5) when $B = 0$. When $A = 0$ and $B \neq 0$ (a case corresponding to a Bingham's rheology) another simple solution can be obtained from (4) by separation of variables as

$$u(t) = \frac{C}{B}(e^{Bt} - 1)$$
$$T = \frac{1}{|B|} \tag{6}$$

Finally, if $A = B = 0$ (a case corresponding to a pure velocity-independent Coulomb's friction) then the straightforward solution is

$$u(t) = C\, t \tag{7}$$

where the characteristic time does not exist anymore, reflecting the property that the flow will infinitely accelerate.

## 3 Application to different rheological models

The quadratic model (3) is used to describe the rheology of debris flows according to a monophasic approximation in some widely used numerical models. In the following sections the analytical solutions (5) and (6) will be used to show how the parameters of (3) affect the flow asymptotic velocity $u_\infty$ and the characteristic time $T$, showing how field observations can

constrain the parameters identification. Furthermore, a sensitivity analysis will provide additional insights on the rheological parameters which most impact the uniform velocity theoretically reached by the flow.

### 3.1 Sobol's global sensitivity analysis

In the following, in order to shed light on the relevance of each parameter on the model output and its interaction with other parameters, a Sobol's global sensitivity analysis is performed (Sobol, 1993) which has been frequently used in environmental

(Saltelli and Annoni, 2010; Estrada and Diaz, 2010) and hydrological modelling (Pappenberger et al., 2008). Assuming a single scalar output denoted $y = f(\boldsymbol{x})$, function of a vector of scalar variables $\boldsymbol{x} = (x_1, x_2, \ldots, x_n)$ possibly grouped into two complementary vectors $(\boldsymbol{u}, \boldsymbol{v})$, the first order and total order indices are respectively defined as (Azzini et al., 2021)

$$
\begin{aligned}
S_u &= \frac{\mathbb{V}[\mathbb{E}[y|\boldsymbol{u}]]}{\mathbb{V}[y]} \\
ST_u &= \frac{\mathbb{V}[\mathbb{E}[y|\boldsymbol{v}]]}{\mathbb{V}[y]}
\end{aligned}
\tag{8}
$$

where, $\mathbb{V}[.]$ stands for the unconditional variance operator (respectively $\mathbb{V}[.|.]$ the conditional variance) and $\mathbb{E}[.]$ stands for the mathematical expectation (respectively $\mathbb{E}[.|.]$ the conditional expectation). The amount of variance relative to the total

variance, called Sobol' sensitivity index, can be attributed either to a single parameter (first order index, $S_u$) or to the interaction of a single parameter with respect to the others (total order index, $ST_u$). The Sobol's indices may vary between 0 and 1. Furthermore, the difference between $S_u$ and $ST_u$ is that the total order index accounts not only for the amount of variance of $y$ explained by the input variables inside $\boldsymbol{u}$ (like $S_u$) but it also contains contributions arising due to the interactions between the variables in $\boldsymbol{u}$ with those in $\boldsymbol{v}$ (Azzini et al., 2021). It can be shown that $ST_u > S_u$ and $S_u + ST_v = 1$, see Azzini et al. (2021)

and Saltelli (2002) for further details. The computation of Sobol' indices could be easily accomplished by a Monte Carlo method, checking all possible parameter combinations, if the function to evaluate is not excessively complicated. However, from the computational point of view, it can be better to evaluate the first and total order indices according to the estimators recently proposed by Azzini et al. (2021), due to their simplicity in implementation. These estimators provide a better performance with respect to the ones originally introduced by Saltelli (2002). For each rheological model evaluation, the set

of parameters has been selected using the latin hypercube sampler (Azzini et al., 2021). Each parameter is assumed to vary uniformly inside the range obtained considering typical values from the literature for each considered rheology. The uniform probability distribution reflects the absence of information about the flow being modelled but, if laboratory or field measurements are available, other parameter distributions can be used. For each model a conservative random sample size of 50 000 parameter sets has been used, repeating the analysis 100 times to ensure the robustness of the Sobol' indices.

## 3.2 O'Brien and Julien's rheology

One of the most used commercial software for the assessment of debris flow hazard is FLO-2D (O'Brien et al., 1993). To model debris flows, FLO-2D uses an ad-hoc rheology proposed by O'Brien and Julien (1988), as

$$S_f = \frac{\tau_y}{\gamma_m\,h} + \frac{K\,\eta\,u}{8\,\gamma_m\,h^2} + \frac{n_t^2 u^2}{h^{4/3}} \tag{9}$$

where $\tau_y$ $[N\,m^{-2}]$ is the yield stress of the granular material, $\gamma_m$ $[N\,m^{-3}]$ is the equivalent fluid specific weight, $K$ $[-]$ is a resistance parameter for laminar flow, $\eta$ $[Pa\ s]$ is the viscosity and $n_t$ $[s\,m^{-1/3}]$ is the turbulent Manning's coefficient. Specific weight, viscosity, yield stress and turbulent Manning's coefficient are functions of the volumetric sediment concentration $C_v$ $[-]$ according to the relations (O'Brien and Julien, 1988)

$$\eta = \alpha_1 e^{\beta_1 C_v}; \quad \tau_y = \alpha_2 e^{\beta_2 C_v}$$
$$n_t = 0.0538\,n\,e^{6.0896\,C_v}; \quad \gamma_m = C_v\,\gamma_s + (1 - C_v)\,\gamma_w \tag{10}$$

where $\alpha_1$ $[Pa\ s], \alpha_2 [Pa], \beta_1 [-]$ and $\beta_2$ $[-]$ are empirical coefficients that could be defined by laboratory experiments (O'Brien and Julien, 1988), $n$ $[s\,m^{-1/3}]$ is the Manning's coefficient while $\gamma_s$ $[N\,m^{-3}]$ and $\gamma_w$ $[N\,m^{-3}]$ are the sediment and water specific weight (the latter assumed fixed in this work) respectively. Accordingly, apart from Manning's coefficient and sediment specific weight, that can be more easily identified, the model is a function of the six parameters $\alpha_1, \beta_1, \alpha_2, \beta_2, K$ and $C_v$. The solution (5) can be used with the following meaning of the coefficients

$$A = -\frac{g\,n_t^2}{h^{4/3}}; \quad B = -\frac{g\,K\,\eta}{8\,\gamma_m\,h^2}; \quad C = g\sin\vartheta - \frac{g\tau_y}{\gamma_m h} \tag{11}$$

and the asymptotic velocity $u_\infty$ is given by

$$u_\infty = \left[ -\frac{K\,\eta}{8\,\gamma_m\,h^2} + \sqrt{\left(\frac{K\,\eta}{8\,\gamma_m h^2}\right)^2 + 4\frac{n_t^2}{h^{4/3}}\left(\sin\vartheta - \frac{\tau_y}{\gamma_m h}\right)} \right]\frac{h^{4/3}}{2n_t^2} \tag{12}$$

whereas the characteristic time is

$$T = \frac{2}{\sqrt{\left(\frac{K\,\eta\,g}{8\,\gamma_m h^2}\right)^2 + 4\frac{g^2\,n_t^2}{h^{4/3}}\left(\sin\vartheta - \frac{\tau_y}{\gamma_m h}\right)}} \tag{13}$$

By inspection of Eq. (12) and (13) one can obtain some interesting insights that can be useful for the calibration of a FLO-2D based model:

- $K$ always appears multiplied by $\eta$ and thus $K$ and $\alpha_1$ are not observable, i.e. their value can't be separately estimated from observations, (e.g. Sorooshian & Gupta, 1983). Accordingly they can be considered as a single parameter.
- The asymptotic velocity is positive if and only if

$$\sin\vartheta - \frac{\tau_y}{\gamma_m h} > 0 \tag{14}$$

that is, among others, a non-linear function of the concentration $C_v$, since $\tau_y$ and $\gamma_m$ depend on the concentration. This condition can also be regarded as one controlling the progressive flow deceleration of the modelled equivalent fluid.

We believe that Eq. (12) and (13) can be useful for the calibration of the numerical model. As an example, Fig. 1 shows the dimensionless analytical solution for the "Aspen Pit 1" parameters set (O'Brien and Julien, 1988), considering typical values $h = 1\,m$ and $\vartheta = 20°$. For this set of values, $u_\infty = 4.85\,m\,s^{-1}$ and $T = 2.89\,s$. Accordingly, $u(T)/u_\infty = 0.86$. The maximum sediment concentration for the flow to develop a uniform velocity is computed by solving Eq. (14), i.e. $C_v = 0.49$; by exceeding this limiting concentration, the flow will slow down and stop.

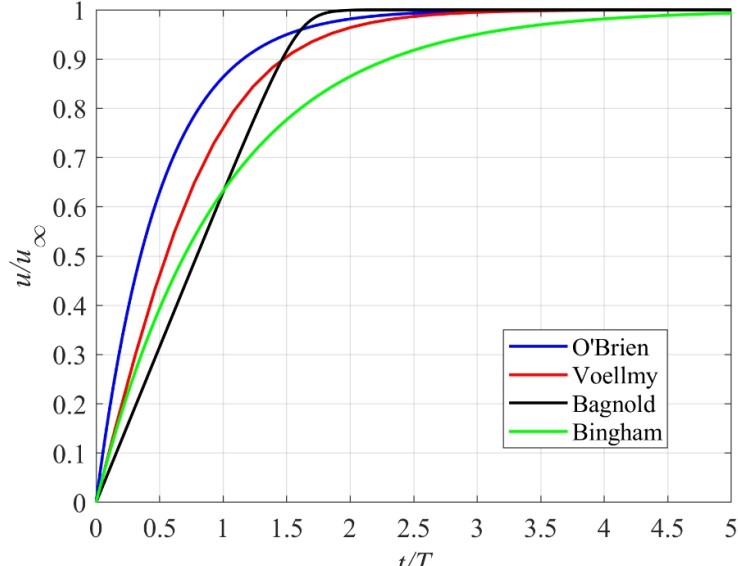

Figure 1. Dimensionless analytical solution for 4 different rheological models. The solution (5) using the O'Brien rheology is shown in blue using the "Aspen Pit 1" parameters set with $\gamma_w = 9\,810\,N\,m^{-3}$ (water specific weight), $\gamma_s = 26\,500\,N\,m^{-3}$ (specific weight of sediments), $n = 0.033\,s\,m^{-1/3}$ and $K = 2285$. The solid red line is the analytical solution (17) for the Voellmy rheology using $\mu = 0.1$ and $\xi = 1000\,m\,s^{-2}$. The solid green line shows the solution computed using Bingham frictional law (21) with $\tau_c = 200\,Pa$, $\gamma_m = 15\,000\,N\,m^{-3}$ and $\eta = 90\,Pa\,s$. Finally the black line is the numerical solution using Bagnold frictional law with $\rho_s = 2\,300\,kg\,m^3$, $\rho_w = 1\,000\,kg\,m^{-3}$, $\phi_d = 35°$, $Y = 40$, $\beta = 1$ and $c_b = 0.65$. In this case, being a numerical solution, there is no formula for the characteristic time, and the time at which the flow reaches 63% of the uniform velocity has been arbitrarily selected as the characteristic time for the normalization. In all cases the debris flow layer $h$ is equal to $1\,m$ and the channel slope is $20°$.

| | $T\,[s]$ | $u_\infty\,[m\,s^{-1}]$ | $u(T)/u_\infty\,[-]$ |
|---|---|---|---|
| O'Brien | 2.89 | 4.85 | 0.86 |
| Voellmy | 6.47 | 15.74 | 0.76 |
| Bagnold | 0.52 | 2.77 | 0.63 |
| Bingham | 5.32 | 16.74 | 0.63 |

Table 1. Numerical summary of the characteristic time $T$, terminal velocity $u_\infty$ and ratio between the velocity for $t = T$, $u(T)$, and the terminal one $u_\infty$ for all rheological models described.

Despite the large number of parameters to be calibrated, the O'Brien rheology is widely used in monophasic numerical models (FLO-2D and HEC RAS) to replicate past debris flow events (e.g. Cesca and D'Agostino, 2008; Lin et al., 2011; Wu et al., 2013; Stancanelli and Foti, 2015; Wang et al., 2024). Calibration of parameters is performed either a posteriori, i.e. using observations of depositional height or maximum velocity, or using rheological investigations (Boniello et al., 2010). In this process one may wonder which parameters have the greatest impact on the simulation results, a type of information that is difficult to know a priori, even when they appear explicitly like in Eq. (12). Although this information can be obtained by performing multiple simulations that encompass the range of variation of each parameter of the friction law, this way remains impractical if one does not use parallelization techniques as done by Zegers et al. (2020). On the other hand, Sobol's analysis quantifies the amount of variance that each parameter contributes to the unconditional variance of the model output,

highlighting which of the parameters have a greater impact on the results. The velocity distributions obtained from random parameter sets used to perform the Sobol's analysis corresponding to each rheological model are reported in Fig. 2. The variation range of the parameters has been obtained by considering typical values from in the literature, as shown in Table 2.

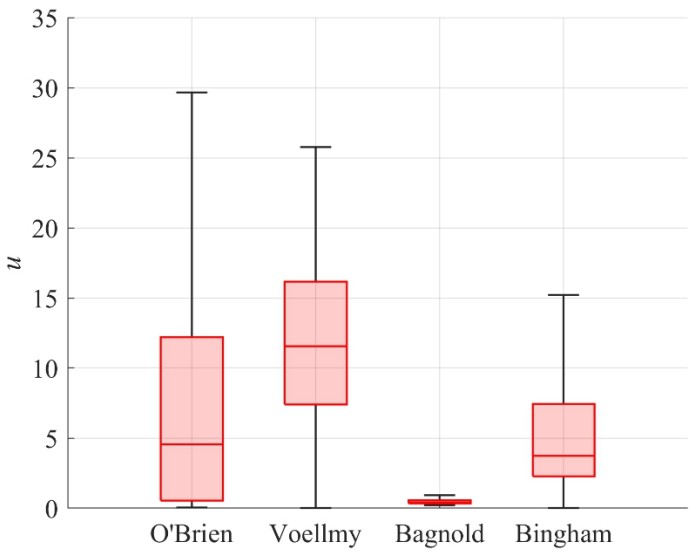

**Figure 2. Uniform velocity boxplot distribution for the 4 rheological models used to perform Sobol sensitivity analysis ($h = 1\ m$, $\vartheta = 20°$). For each graph the solid line represents the median value across the whole distribution, the rectangular box covers the range from the lower quartile (25%) to the upper quartile (75%) and the whiskers show the non-outlier maximum (the maximum data value which is not an outlier) and minimum (the minimum data value which is not an outlier). Here a point is classified as an outlier if it lies more than $1.5$ times the interquartile range away from the top or bottom of the red-shaded rectangle.**

| Parameter | Minimum | Maximum | Units | Reference |
|:---:|:---:|:---:|:---:|:---:|
| $\alpha_1$ | $3 \cdot 10^{-5}$ | $6.4 \cdot 10^{-3}$ | $Pa\ s$ | Sosio et al., (2007) |
| $\beta_1$ | 6.2 | 33.1 | – | Zegers et al., (2020) |
| $\alpha_2$ | $7.1 \cdot 10^{-5}$ | 0.0181 | $Pa$ | Zegers et al., (2020) |
| $\beta_2$ | 16.9 | 29.8 | – | O'Brien and Julien, (1988) |
| $C_v$ | 0.2 | 0.55 | – | Zegers et al., (2020) |
| $n$ | 0.01 | 0.2 | $s\ m^{-1/3}$ | Zegers et al., (2020) |
| $K$ | 24 | 2285 | – | O'Brien and Garcia, (2009) |
| $\alpha_k$ | $7.2 \cdot 10^{-4}$ | 14.62 | $Pa\ s$ | O'Brien & Garcia, (2009); Sosio et al., (2007) |
| $\gamma_s$ | 18 | 25 | $kN\ m^{-3}$ | Iverson, (1997) |

**Table 2. Variation range for the parameters tested in the Sobol analysis.**

By performing the computations, the first-order Sobol' indices (shown in Fig. 3) highlight that the most influential parameters are $C_v$, $\beta_1$ and $n$, in order of importance. $C_v$ is able, by itself, to explain around 35% of the variance of the terminal velocity, as one could expect considering that $C_v$ influences all the terms in Eq. (12) whereas the other parameters affect just a single term of the equations. High model sensitivity to $\beta_1$ (almost 20%) confirms other findings in literature (Zegers et al., 2020). The sensitivity on the Manning's coefficient (around 13%) reflects the importance of surface roughness and associated turbulent friction. This is not in line with the findings of Zegers et al. (2020), in which model results are insensitive to Manning's roughness coefficient. This can be explained considering that the reference velocity considered by Zegers et al. (2020), which appears quadratically in the term where the Manning's coefficient appears, is around $1\ m\ s^{-1}$, in contrast with the wider range of velocities considered in this study (see Fig. 2). Parameters $\alpha_k$, $\alpha_2$, $\beta_2$ and $\gamma_s$ appear to be less important in terms of the terminal velocity.

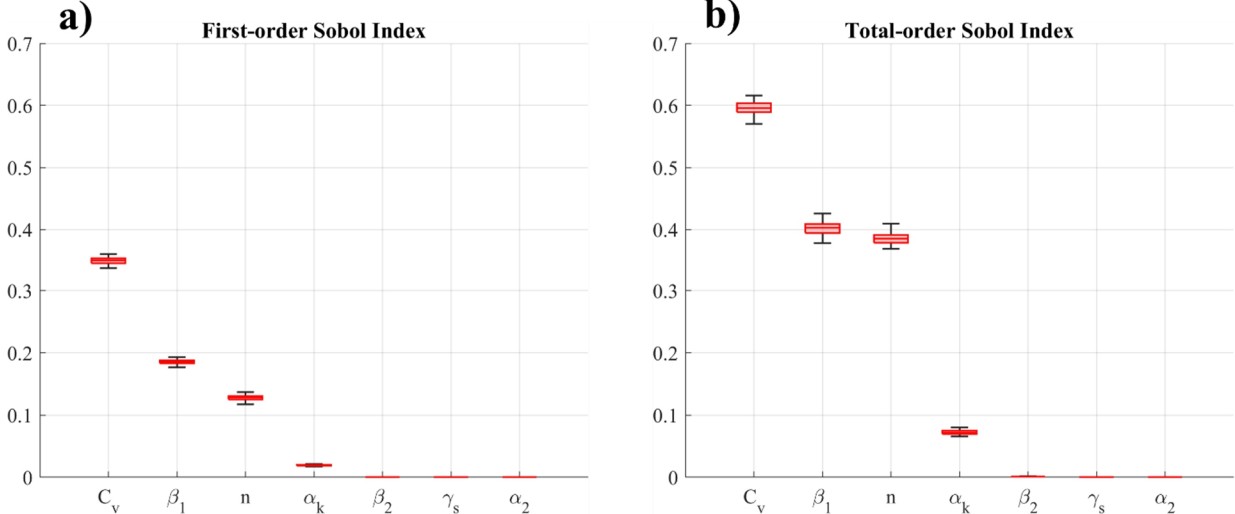

**Figure 3.** Boxplot of the first-order Sobol' index (a) and total order index (b) for the parameters of the O'Brien rheology, ordered in terms of importance.

**Normalized $C_v$ First-order Sobol Index**

|  | h = 0.5 m | h = 1 m | h = 2 m |
|---|---|---|---|
| $\vartheta = 10°$ | 0.76 | 0.84 | 0.89 |
| $\vartheta = 20°$ | 0.94 | 1.02 | 1.06 |
| $\vartheta = 30°$ | 1.11 | 1.17 | 1.21 |

**Normalized $C_v$ Total-order Sobol Index**

|  | h = 0.5 m | h = 1 m | h = 2 m |
|---|---|---|---|
| $\vartheta = 10°$ | 0.95 | 0.95 | 0.96 |
| $\vartheta = 20°$ | 1.00 | 1.00 | 1.00 |
| $\vartheta = 30°$ | 1.04 | 1.04 | 1.05 |

**Normalized $\beta_1$ First-order Sobol Index**

|  | h = 0.5 m | h = 1 m | h = 2 m |
|---|---|---|---|
| $\vartheta = 10°$ | 1.39 | 1.27 | 1.21 |
| $\vartheta = 20°$ | 1.09 | 0.98 | 0.92 |
| $\vartheta = 30°$ | 0.80 | 0.70 | 0.64 |

**Normalized $\beta_1$ Total-order Sobol Index**

|  | h = 0.5 m | h = 1 m | h = 2 m |
|---|---|---|---|
| $\vartheta = 10°$ | 1.42 | 1.27 | 1.19 |
| $\vartheta = 20°$ | 1.11 | 0.97 | 0.91 |
| $\vartheta = 30°$ | 0.81 | 0.69 | 0.63 |

**Normalized n First-order Sobol Index**

|  | h = 0.5 m | h = 1 m | h = 2 m |
|---|---|---|---|
| $\vartheta = 10°$ | 0.61 | 0.71 | 0.76 |
| $\vartheta = 20°$ | 0.90 | 1.01 | 1.07 |
| $\vartheta = 30°$ | 1.22 | 1.33 | 1.39 |

**Normalized n Total-order Sobol Index**

|  | h = 0.5 m | h = 1 m | h = 2 m |
|---|---|---|---|
| $\vartheta = 10°$ | 0.89 | 0.91 | 0.91 |
| $\vartheta = 20°$ | 1.00 | 1.01 | 1.01 |
| $\vartheta = 30°$ | 1.08 | 1.09 | 1.10 |

**Figure 4.** First-order (left) and total-order (right) Sobol index for the parameters $C_v, \beta_1$ and $n$ as a function of different values of $\vartheta$ and $h$. Each entry of the matrix represents the first- (total-) order Sobol index computed using the corresponding $(h, \vartheta)$ couple, then divided by the average index over the whole matrix. The average first-order (total-order) Sobol indices for $C_v, \beta_1$ and $n$ are $0.34$ $(0.6)$, $0.19$ $(0.41)$ and $0.13$ $(0.38)$ respectively.

Globally, the sum of the first-order Sobol' indices can explain 68% of the variance of the model meaning that the remainder of the variance is due to the nonlinearity of the model, i.e. the variance generated by a combination of parameters. Total-order indices reveals the importance of the interactions between $C_v, \beta_1$ and $n$. As observed above, the results of the sensitivity analysis are valid for a given couple of the fluid depth $h$ and channel slope $\vartheta$: therefore the following analysis has been repeated to explore the variation of the Sobol' indices as a function of $(h, \vartheta)$. Fig. 4 shows the Sobol' indices for $C_v, \beta_1$ and $n$,

normalized by dividing each index by its average value across the whole matrix. The colour shading shows the growing direction for each index. As one can observe, the first-order Sobol' index of $C_v$ and $n$ is a growing function of $h$ and $\vartheta$ whereas their total-order index is mostly a growing function of $\vartheta$. Considering $\beta_1$, both the first-order and total-order Sobol' indices show that for growing values of $h$ and $\vartheta$ the sensitivity of the uniform velocity decreases, suggesting the growing importance of viscosity for small fluid depths or channel slope.

**3.3 Voellmy's rheology**

The Voellmy's rheology (Voellmy, 1955) splits the total basal friction into a velocity independent dry-Coulomb term proportional to the normal stress at the flow bottom and a velocity dependent friction (Salm, 1993). This simple friction law is implemented in several numerical software, e.g. RAMMS (Christen et al., 2010), HEC-RAS Mud and Debris Flow (US Army Corps of Engineers, 2023) and FLATModel (Medina et al., 2007). The Voellmy's friction law is given by

$$S_f = \mu \cos \vartheta + \frac{u^2}{\xi h} \qquad (15)$$

where $\mu \, [-]$ is a Coulomb friction coefficient and $\xi \, [m \, s^{-2}]$ is a turbulent friction coefficient. The analytical solution to Eq. (2) is given by Eq. (5) using the following coefficients

$$A = -\frac{g}{\xi h}; \quad B = 0; \quad C = g \sin \vartheta - g \mu \cos \vartheta \qquad (16)$$

obtaining the solution suggested by Voellmy (1955) and Hergarten and Robl (2015)

$$u(t) = u_\infty \tanh\left(\frac{t}{T}\right)$$
$$u_\infty = \sqrt{\xi h (\sin \vartheta - \mu \cos \vartheta)}; \quad T = \frac{\xi h}{g \, u_\infty} \qquad (17)$$

Eq. (17) is a simple and interesting piece of information to evaluate the appropriateness of the parameters used to propagate the debris flow being modelled. Once again by inspection of the terminal velocity of Eq. (17) one can conclude that the flow

can approach a uniform condition only when $\tan \vartheta > \mu$, meaning that the slope of the channel must be higher than the friction coefficient. Assuming invariance of the parameters during the motion, this condition can be used to calibrate the friction parameter $\mu$ that must be upper limited by the local slope angle in the inception area and lower limited by the local slope angle where the debris flow has stopped its propagation. The analytical solution (17) is shown in Fig. 1 in correspondence of the parameters set $\mu = 0.1, \xi = 1\,000 \, m \, s^{-2}, h = 1 \, m$ and $\vartheta = 20°$. After a characteristic time $(6.47 \, s)$ the flow approaches a

velocity which is 76% of the terminal one $(15.74 \, m \, s^{-1})$. Performing the Sobol sensitivity analysis described above yields the results reported in Table 3, where the first line is the variation range explored in the analysis and $h = 1 \, m$ and $\vartheta = 20°$. The indices highlight a balanced sensitivity of the uniform velocity to both parameters, with $\mu$ scoring slightly higher both in terms of the first- and total-order indices.

|  | $\xi$ | $\mu$ |
|---|---|---|
| Range | $50 - 2\,000\ m\ s^{-2}$ | $0.01 - 0.4$ |
| First-order index | 0.40 | 0.55 |
| Total-order index | 0.44 | 0.60 |

**Table 3. Summary of the Sobol sensitivity analysis obtained with a fluid depth of 1 $m$ and a channel slope of 20°. The parameter's range is taken from RAMMS user manual.**

Repeating the analysis for different values of $h$ and $\vartheta$ shows little variation in terms of the computed sensitivity indices (both first- and total-order ones); therefore the indices reported in Table 3 are representative of the whole spectrum of variation of $h$ and $\vartheta$.

## 3.4 Bagnold's rheology

Bagnold's rheology (1954) modified by Takahashi (1978) on the basis of experimental data and valid for stony-type debris flow is used, for instance, inside the TRENT2D software (Rosatti and Begnudelli, 2013):

$$S_f = \frac{25}{4}\frac{\rho_s}{\rho_w}\sin\phi_d\frac{\lambda^2}{Y^2(1 + C_v\,\Delta_s)\ g\ h}u^2$$

$$\lambda = \left[(c_b/C_v)^{1/3} - 1\right]^{-1}; \quad \Delta_s = \frac{\rho_s - \rho_w}{\rho_w}; \quad C_v = c_b\beta\frac{u^2}{gh}$$

(18)

where $\rho_s\ [kg\ m^{-3}]$ is the density of the sediments, $\rho_w\ [kg\ m^{-3}]$ is the density of water, $\phi_d\ [°]$ is the friction angle of the sediments, $Y\ [-]$ is the submergence parameter, $\Delta_s$ is the submerged relative density of the sediments, $c_b\ [-]$ is the maximum concentration of the mixture in static conditions, $C_v\ [-]$ is the concentration of the mixture and $\beta$ is a transport parameter to be calibrated. Although the submergence parameter depends on other factors, following Rosatti and Begnudelli (2013) we kept it as a calibration parameter. Due to the function $\lambda$, which depends non-linearly on the velocity, a simple analytical expression of the solution of (2) is no longer obtainable. Similarly, a simple closed-form expression for the uniform velocity can't be obtained, due to the non-integer exponents present inside Eq. (18). Accordingly, to compare the results with previous models, we solved Eq. (2) numerically, by using a finite difference method. The solution is shown in Fig. 1, for $\rho_s = 2\,300\ kg\ m^{-3}, \rho_w = 1\,000\ kg\ m^{-3}, \phi_d = 35°, Y = 40, \beta = 1, c_b = 0.65,\ h = 1\ m$ and $\vartheta = 20°$. For this combination, the uniform velocity is $2.77\ m\ s^{-1}$ and the characteristic time is $0.52\ s$. Being a numerical solution, no expression for the characteristic time is available, and the time when the flow reaches 63% of the uniform velocity has been arbitrarily regarded as the characteristic time, in analogy with an exponential decay. Contrary to the rheological models studied so far, Bagnold's model provides a uniform velocity for every choice of the parameters, when $\vartheta > 0$ . The Sobol's analysis has been repeated for the Bagnold's rheology exploring the parameters range shown in Table 4, taken from literature (e.g. Stancanelli and Foti, 2015).

|  | $\beta$ | $Y$ | $\phi_d$ | $\rho_s$ | $c_b$ |
|---|---|---|---|---|---|
| Range | $0.1 - 100$ | $1 - 50$ | $20 - 45°$ | $1\,800 - 2\,500\ kg\ m^{-3}$ | $0.4 - 0.65$ |
| First-order index | 0.95 | 0.01 | $9 \cdot 10^{-5}$ | $3 \cdot 10^{-6}$ | $3 \cdot 10^{-6}$ |
| Total-order index | 0.98 | 0.04 | $5 \cdot 10^{-4}$ | $3 \cdot 10^{-5}$ | $10^{-5}$ |

**Table 4. Summary of the Sobol sensitivity analysis obtained with a fluid depth of 1 $m$ and a channel slope of 20°.**

The indices of Table 4 show that the most important parameter is $\beta$, which by itself is able to explain more than 90% of the variance, while the others show limited influence on the uniform velocity. Repeating the analysis for different values of $h$ and $\vartheta$ shows little to no variation in terms of the computed sensitivity indices (both first- and total-order ones); therefore the indices shown in Table 4 are representative for the whole spectrum of variation of $h$ and $\vartheta$. Furthermore, considering the uniform

velocity distributions showed in Fig. 2 for the different rheological models it is possible to notice that the Bagnold rheology is characterized by a narrow field of variation of the velocity $(0 - 3 \, m \, s^{-1})$ with respect to the other rheologies. This finding is

in line with other comparative works in literature (e.g. Stancanelli & Foti, 2015) where it is observed that FLO-2D (O'Brien rheology) can predict velocities during a general simulation which are considerably higher $(1.5 - 20 \, m \, s^{-1})$ compared with the TRENT2D (Bagnold rheology) ones $(1 - 2 \, m \, s^{-1})$ on the same simulation.

### 3.5 Bingham's rheology

The Bingham's (1922) rheology is another commonly adopted model for debris flow propagation (Malet et al. 2005; Begueria

et al., 2009), implemented, for instance, inside the TELEMAC-2D software (TELEMAC-2D User Manual, 2017). The main assumption in the Bingham model of laminar flow regime of viscoplastic materials with a constant value of yield strength and viscosity is especially valid in flows with high fine fraction (Abraham et al., 2022). Using the shear stress equation reported in Coussot (1997), a simplified Bingham expression of shear stress has been used to derive the value of the friction slope term, as proposed by (Abraham et al., 2022):

$$S_f = \frac{3 \, \tau_c}{2 \, \gamma_m \, h \cos\vartheta} + \frac{3 \, \eta \, u}{\gamma_m h^2 \cos\vartheta} \tag{19}$$

where $\tau_c \, [Pa]$ is the constant yield strength due to cohesion of soil in the static case, $\eta \, [Pa \, s]$ is the viscosity parameter and $\gamma_m \, [N \, m^{-3}]$ is the specific weight of the mixture. Using the following coefficients

$$A = 0; \quad B = -\frac{3 \, \eta \, g}{\gamma_m h^2 \cos\vartheta}; \quad C = g \sin\vartheta - \frac{3 \, \tau_c \, g}{2 \, \gamma_m \, h \cos\vartheta} \tag{20}$$

from Eq. (6) one can obtain the analytical solution

$$u(t) = u_\infty \left( e^{-\frac{3 \, \eta \, g}{\gamma_m h^2 \cos\vartheta} t} - 1 \right)$$

$$u_\infty = \frac{\gamma_m h^2 \cos\vartheta}{3 \, \eta} \left( \sin\vartheta - \frac{3 \, \tau_c}{2 \, \gamma_m \, h \cos\vartheta} \right) \tag{21}$$

$$T = \frac{\gamma_m h^2 \cos\vartheta}{3 \, \eta \, g}$$

Finally, the flow will approach a terminal velocity if the quantity inside the brackets of Eq. (21) is positive, i.e. if

$$\sin\vartheta \cos\vartheta > \frac{3 \, \tau_c}{2 \, \gamma_m \, h} \tag{22}$$

providing a simple criterion to understand flow deceleration. Fig. 1 plots the analytical solution Eq. (21) using $\tau_c =$

$200 \, Pa, \eta = 90 \, Pa \, s, \gamma_m = 15\,000 \, N \, m^{-3}, h = 1 \, m$ and $\vartheta = 20°$. After a characteristic time of $5.32 \, s$ the flow reaches a velocity of $16.74 \, m \, s^{-1}$ which is 63% of the terminal velocity. Interestingly, the parameter $\tau_c$ does not influence the characteristic time of the flow, but only the terminal velocity. Table 5 shows the results obtained by performing Sobol's sensitivity analysis in the range taken from Phillips and Davies (1991), properly extended due to lack of specific information regarding the simplified formulation adopted and using $h = 1 \, m$ and $\vartheta = 20°$. The indices highlight a high sensitivity of the

uniform velocity to the viscosity $\eta$, while the other parameters show negligible influence on the model's output. This finding is consistent with what already found using the O'Brien rheology, i.e. that parameters that control viscosity strongly influences the uniform velocity.

|  | $\eta$ | $\tau$ | $\gamma_m$ |
|---|---|---|---|
| Range | $1 - 4 \cdot 10^3 \, Pa \, s$ | $0.1 - 5 \cdot 10^4 \, N \, m^{-2}$ | $16 - 23 \, kN \, m^{-3}$ |
| First-order index | 0.82 | $3 \cdot 10^{-3}$ | $10^{-4}$ |
| Total-order index | 0.99 | 0.16 | 0.014 |

**Table 5. Summary of the Sobol sensitivity analysis, assuming a fluid depth of $1 \, m$ and a channel slope of $20°$.**

Repeating the analysis for different values of $h$ and $\vartheta$ shows little to no variation in terms of the computed sensitivity indices (both first- and total-order ones). Accordingly the indices reported in Table 5 are representative of the whole spectrum of variation of $h$ and $\vartheta$.

## 4 Application and discussion

The proposed analytical solutions can be applied to simplify the calibration of parameters when some insights on the expected debris flow velocity are available, as shown in the following with reference to the debris flow that occurred in the Blé basin in Valle Camonica valley, in the Central Italian Alps (Lombardia Region). On August 16, 2021, a debris flow was triggered by an intense rainfall of 40 mm in 1 hour, caused by a localized storm cell moving west to east (Berti et al., 2023). The event mobilized a total volume of about 60 000 $m^3$ and travelled along the channel for about 2 kilometres before covering a local

road with debris and boulders. A monitoring station (Berti et al., 2023) provided video footage of the debris flow through a known cross-section.

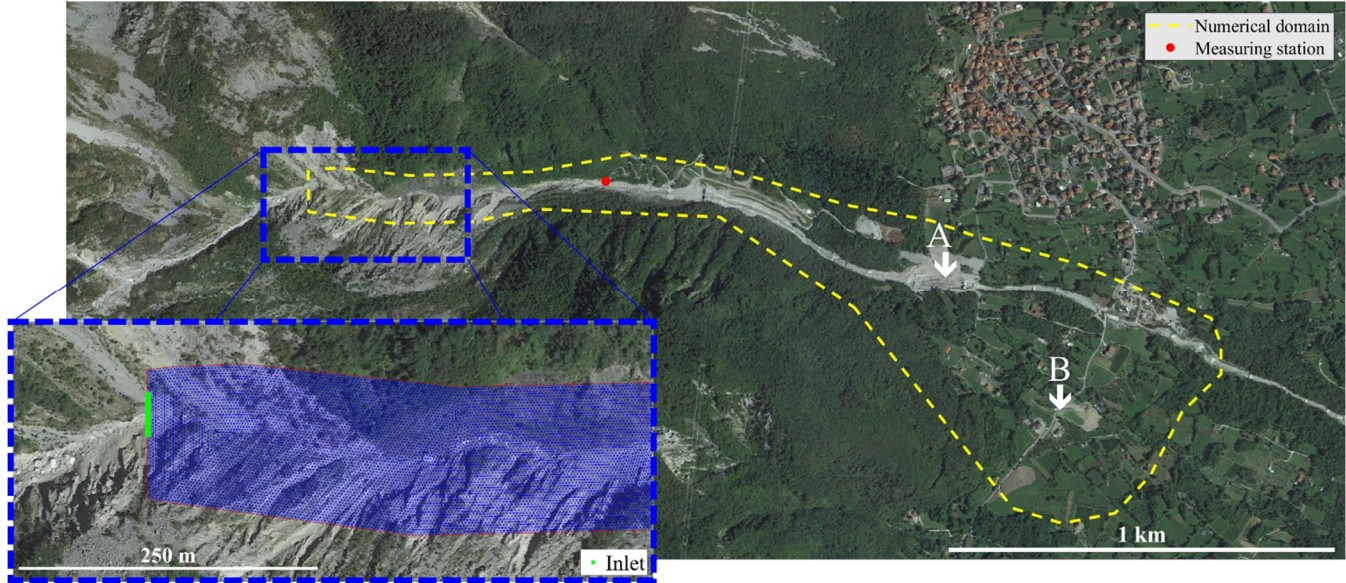

**Figure 5. Map view of the study area. The yellow dashed line highlights the computational domain used in this numerical simulation, the red dot illustrates the position of the recording station while the dashed blue line shows an enlargement of the inlet cross section**

**(depicted in green) as well as the unstructured mesh adopted for the simulation. Locations A and B indicate the main areas where the debris flow deposited, (© Google Earth 2019).**

Using video footage and cross-section geometry, Berti et al. (2023) combined two free Matlab tools (PIVlab, Thielicke and René, 2021 and RiveR, Patalano et al., 2017) in order to estimate the peak flow velocity and discharge. The obtained peak flow velocity and discharge are $4.4 \, m \, s^{-1}$ and $224 \, m^3 \, s^{-1}$ respectively during the first surge and $5.4 \, m \, s^{-1}$ and $227 \, m^3 \, s^{-1}$

during the second one (Berti et al., 2023). Based on the volume of the event and peak discharge, the reconstructed solid hydrograph shown in Fig. 6 was adopted as a boundary condition. The two-peak structure of the hydrograph was obtained by fixing the peaks (both assumed of $227 \, m^3 \, s^{-1}$ for simplicity) as well as the time lag between the first surge and the second

one (about 2 minutes), with the constrain on the overall duration of the event of about 10 minutes. The Finite Volume SWE-based numerical scheme presented in Bonomelli et al. (2023), adapted for mountain area applications (Bonomelli and Pilotti, 2023; Bonomelli, 2024) was used to propagate the debris flow along the main channel of the catchment, with an unstructured triangular mesh consisting of 194 151 elements with an average distance of $1.63\ m$. The resolution of the available Digital Elevation Model (DEM) was $2\ m$. To show how the proposed methodology can be used to guide the calibration process, in the following we will use the two most used rheological models, i.e. Voellmy and O'Brien, to replicate the debris flow occurred in the Blé catchment.

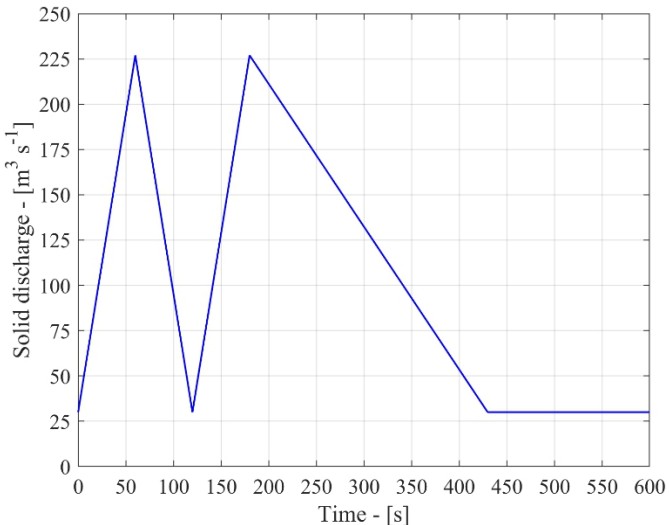

**Figure 6. Solid discharge hydrograph used as a boundary condition for the Blé event. Two discharge peaks were observed during the event, approximately two minutes apart. This information, together with the total volume estimate provided, allowed to fix the shape of the hydrograph. A base flow of $30\ m^3\ s^{-1}$ was assumed after the analysis of the available video footage.**

**4.1 Calibration of the Voellmy model**

The Voellmy rheological model uses only two parameters: $\mu$ and $\xi$: the first one is usually estimated using the local average slope in the main deposition area of the debris flow (locations A and B of Fig. 5), which has been evaluated to be around $14°$ based on the available DEM: accordingly, $\mu = 0.249$. Eq. (17) provides a first guess estimate for $\xi$, using the cross section geometry where the velocity was measured. Setting $u_\infty = 5.4\ m\ s^{-1}$, $h_{average} \approx 2\ m$ (Berti et al., 2023), $\mu = 0.249$ and computing the average slope at the measuring station, i.e. $\vartheta \approx 30°$, one can obtain a first guess for $\xi$

$$\xi = \frac{u_\infty^2}{h\ (\sin\vartheta - \mu\cos\vartheta)} \tag{23}$$

of about $52\ m\ s^{-2}$. The corresponding characteristic time, Eq. (17), is about $2\ s$, confirming that the debris flow rapidly adjusts itself to the local slope and the idea of using a normal flow equation is reasonably grounded. Having determined two first-guess values for the rheological parameters, the numerical simulation can be performed to assess whether the depositional depths and extent reasonably match the observed ones. The stopping criteria for propagation adopted by RAMMS (RAMMS manual, 2022) was implemented, which happened after 17 minutes of simulated time. Fig. 7 shows the simulated deposition map compared to the extent surveyed after the event (see the supplementary material for some photos of the event). The observed qualitative match is good and could be further improved to better reproduce the main characteristics of the event by exploring a neighbourhood of the two identified parameters.

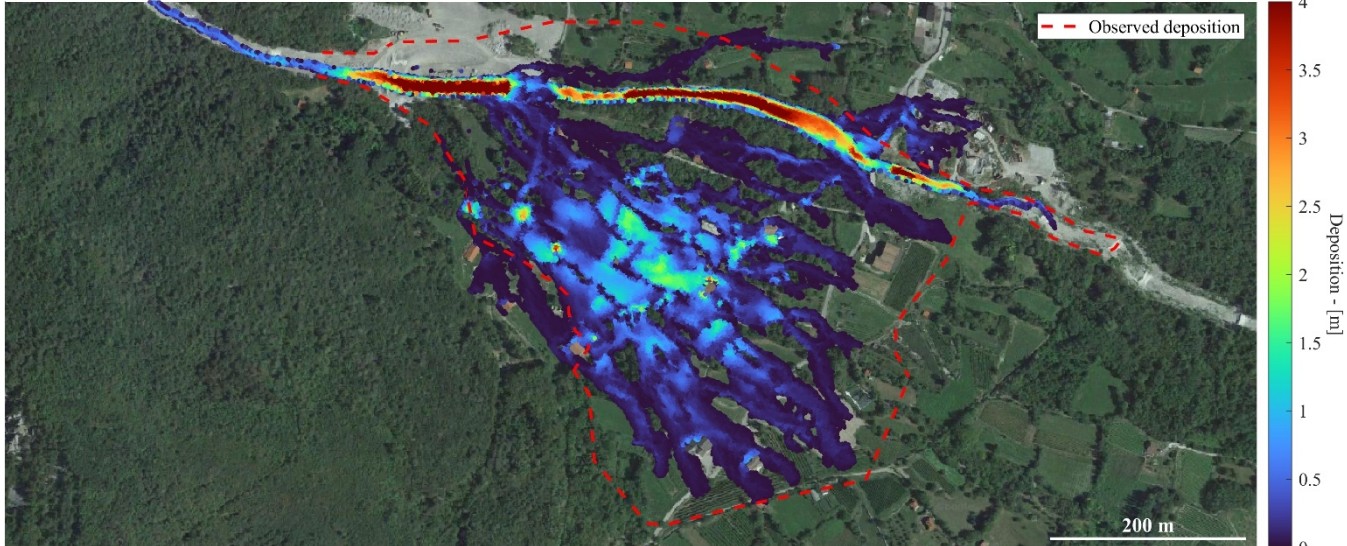

**Figure 7. Deposition map at the end of the simulation using the Voellmy model with $\mu = 0.249$ and $\xi = 52\ m\ s^{-2}$ (© Google Earth 2019).**

Fig. 8 shows the computed velocity profile at the measuring station in correspondence of the peak during the simulation. Quite interestingly, the identified $\xi$ value, although obtained by a steady state formula, provides velocity during the transient process which are in the order of $5 - 6\ m\ s^{-1}$, confirming the role of the small characteristic time and the effectiveness of the simple analytical law adopted. The same figure also shows the higher velocity profile that would be obtained if the same input hydrograph were routed with the same $\mu$ but with a blind initial guess of $\xi$, selected in the middle of its documented range of variation.

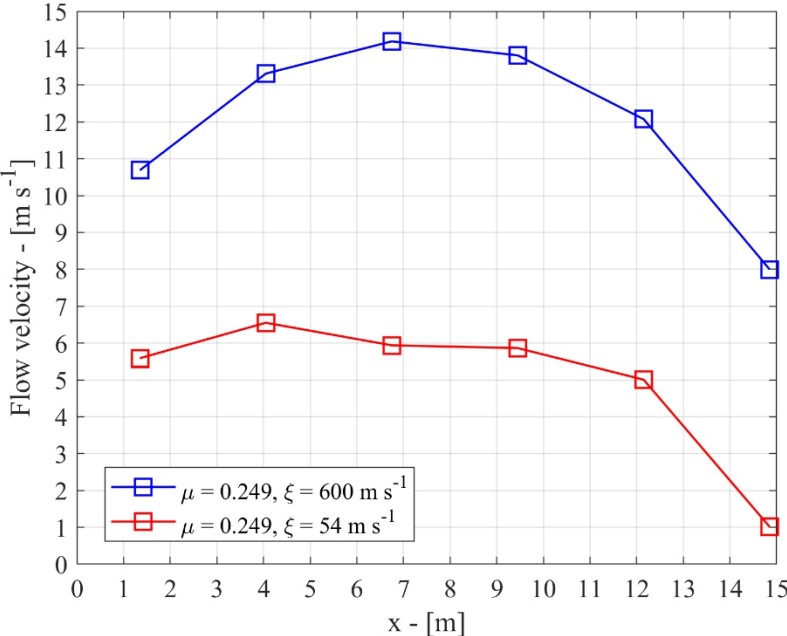

**Figure 8. Peak flow velocity module at the measuring cross-section. The two profiles are obtained with the same $\mu$ value but different $\xi$. For blue squares, $\xi$ was selected in the middle of its range of admissible values, while for red squares, the value of $\xi$ was obtained using the proposed approach exploiting the available velocity measurement.**

Accordingly, the proposed approach provides an effective initial guess for the rheological parameters of the Voellmy's model if a velocity measurement is available at a known cross-section, showing how a simple analytical law can speed up the calibration process. Finally, it is important to observe that another interesting use of Eq. (17) is as a uniform flow relation for the implementation of the inlet (or outlet) boundary condition (Hou et al., 2015) inside a numerical model. For instance, by

rearranging Eq. (17) for the Voellmy rheological model one can easily obtain the incoming discharge as a function of the fluid depth

$$Q(h) = L\, h^{3/2} \sqrt{\xi\,(\sin\vartheta - \mu\cos\vartheta)} \tag{24}$$

where $L$ is the inlet cross section width and $\vartheta$ is the local channel inclination, which must exceed $\mu$ for the flow to occur.

## 4.2 Calibration of the O'Brien model

The second model considered is the widely used O'Brien model. Considering its multiparametric nature, Sobol's analysis suggests that the most influential parameters in terms of the terminal velocity are $C_v$, $\beta_1$ and $n$. The Manning's coefficient is a parameter frequently used in flood modelling therefore its value can be estimated according to the tabulated values existing in literature (e.g. Chow, 1959) and finding the best match by visual inspection. On this basis $n = 0.1\, s\, m^{-1/3}$ was selected for this simulation. The other parameters are less influential in terms of the terminal velocity and can be fixed around their default

values without significantly affecting the computed velocities. Contrary to the Voellmy model, there is no direct way to use the local average slope of the main deposition areas to fix a particular parameter inside the rheological model. Eq. (14) regulates flow deceleration, but the presence of the fluid depth inside the relationship states that the flow will slow down depending on the fluid depth at which it is propagating, thus making it harder to obtain a proper constraint on the rheological parameters that mostly influence deposition. First of all, in Fig. 9 we show the deposition map obtained by running a numerical simulation in

a "blind" way, by using a standard rheological set, i.e. "Aspen Pit 1" (O'Brien & Julien, 1988), without any information to guide the calibration process.

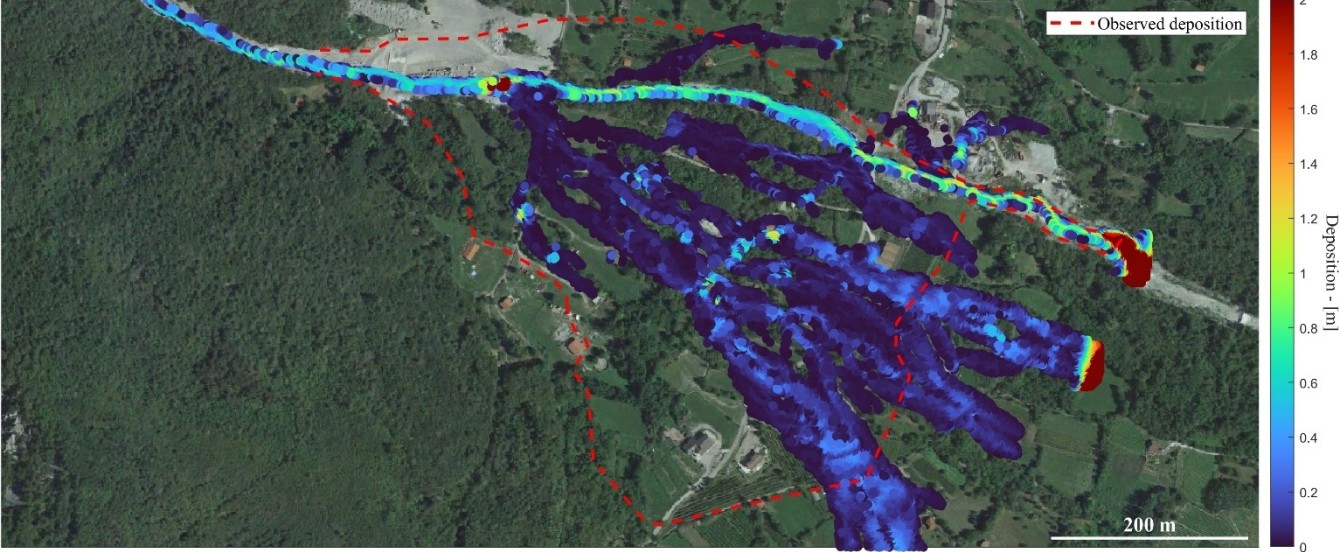

**Figure 9. Deposition map at the end of the simulation using the O'Brien model with the Aspen pit 1 parameters. $C_v = 0.313$, $\alpha_1 = 0.0036\, Pa\, s$, $\alpha_2 = 0.0181\, Pa$, $\beta_1 = 22.1$, $\beta_2 = 25.7$, $\gamma_m = 26.5\, kN\, m^{-3}$, $n = 0.1\, s\, m^{-1/3}$ and $K = 2285$ (© Google Earth**
**2019).**

As it can be noticed the debris flow is too fluid and flows out of the domain, contrary to field observations. Furthermore, Fig. 12 shows the velocity profile computed at the recording station in correspondence of the second peak discharge, which is unrealistically high (around $10\, m\, s^{-1}$) when compared with the recordings ($5.4\, m\, s^{-1}$). This is not surprising considering that no calibration has been performed other than fixing the Manning's coefficient. As a following step, we propose to calibrate

the remaining parameters suggested by the Sobol's analysis by setting the terminal velocity showed in Eq. (12) at the monitoring station equal to the recorded one. Moreover, to make a direct comparison with the previous Voellmy example, we set the characteristic time in Eq. (13) equal to the characteristic time computed during the calibration of the Voellmy rheology,

i.e. $T_{Voellmy} \approx 2\ s$ . This leads to a nonlinear system of two equations showed in Eq. (25) that can be solved numerically to find the parameter values,

$$
\begin{cases}
T_{Voellmy} = \dfrac{2}{\sqrt{\left(\dfrac{K\,\eta\,g}{8\,\gamma_m h^2}\right)^2 + 4\dfrac{g^2\,n_t^2}{h^{4/3}}\left(\sin\vartheta - \dfrac{\tau_y}{\gamma_m h}\right)}} \\[2em]
u_{station} = \left[-\dfrac{K\,\eta}{8\,\gamma_m\,h^2} + \sqrt{\left(\dfrac{K\,\eta}{8\,\gamma_m h^2}\right)^2 + 4\dfrac{n_t^2}{h^{4/3}}\left(\sin\vartheta - \dfrac{\tau_y}{\gamma_m h}\right)}\right]\dfrac{h^{4/3}}{2n_t^2}
\end{cases}
\tag{25}
$$

and whose solution is $C_v = 0.423$ and $\beta_1 = 20.5$, which are then used to propagate the debris flow along the main channel using the same boundary condition and mesh of the previous simulation. By doing this, a great improvement is obtained on the velocity profile computed at the station (see Fig. 12, around $6.4\ m\ s^{-1}$), that is now much closer to the observed one ($5.4\ m\ s^{-1}$) with respect to the previous simulation. However, the computed final deposition (see Fig. 10) shows only a limited improvement because the occurred debris flow does not flow out of the flooded area. Actually, one can observe that there is not a compelling reason to impose the same characteristic time obtained for the Voellmy rheology, as we did for comparison's sake in the system (25). Considering that deposition is controlled by Eq. (14) in the O'Brien model, one can improve the pattern of the depositional area by changing the involved parameters, i.e. either $\alpha_2$ or $\beta_2$, which, in turn, do not affect the terminal velocity significantly according to the Sobol's analysis. For instance, Fig. 11 shows the deposition obtained by setting $\alpha_2 = 0.001\ Pa$, which improves the previous simulation without altering significantly the velocity profile computed at the recording station (see Fig. 12). This example shows how it is possible to get to the neighbourhood of an acceptable and physical set of parameters in a few trials with the proposed approach.

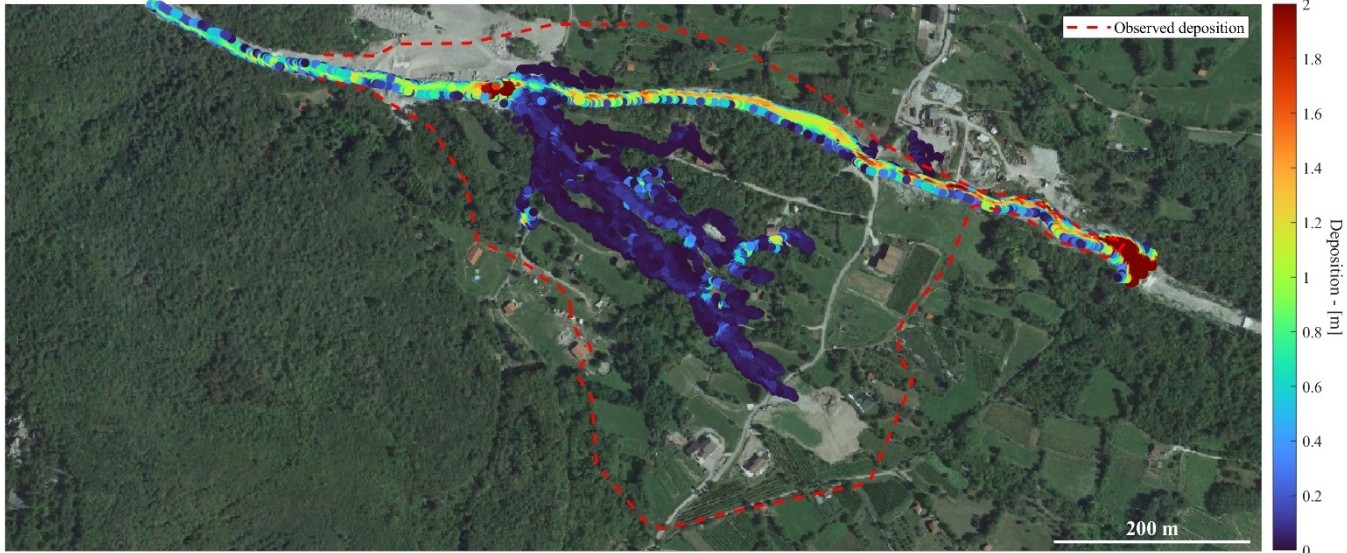

**Figure 10. Deposition map at the end of the simulation using the O'Brien model with $C_v = 0.423, \alpha_1 = 0.0036\ Pa\ s, \alpha_2 = 0.0181\ Pa, \beta_1 = 20.5, \beta_2 = 25.7, \gamma_m = 26.5\ kN\ m^{-3}, n = 0.1\ s\ m^{-1/3}$ and $K = 2285$ (© Google Earth 2019).**

It may be worthwhile to observe that with the O'Brien model the debris flow propagates out of the domain along the main channel regardless of the rheological set adopted, contrary to field observations. This behaviour is known and can be explained on the basis of Eq. (14) that shows that the critical yield stress to stop the motion increases with the flow depth, whereas in the Voellmy rheology this behaviour is mitigated by the constant friction term (see Eq. 15) which is independent from the fluid depth. Therefore, in the Voellmy rheology, a debris flow which encounters a slope which is less than its internal friction angle $\mu$ will tend to slow down and deposit for any fluid depth. This is not true for the O'Brien rheology, where friction decreases with the flow depth, contributing to a more fluid-like behaviour.

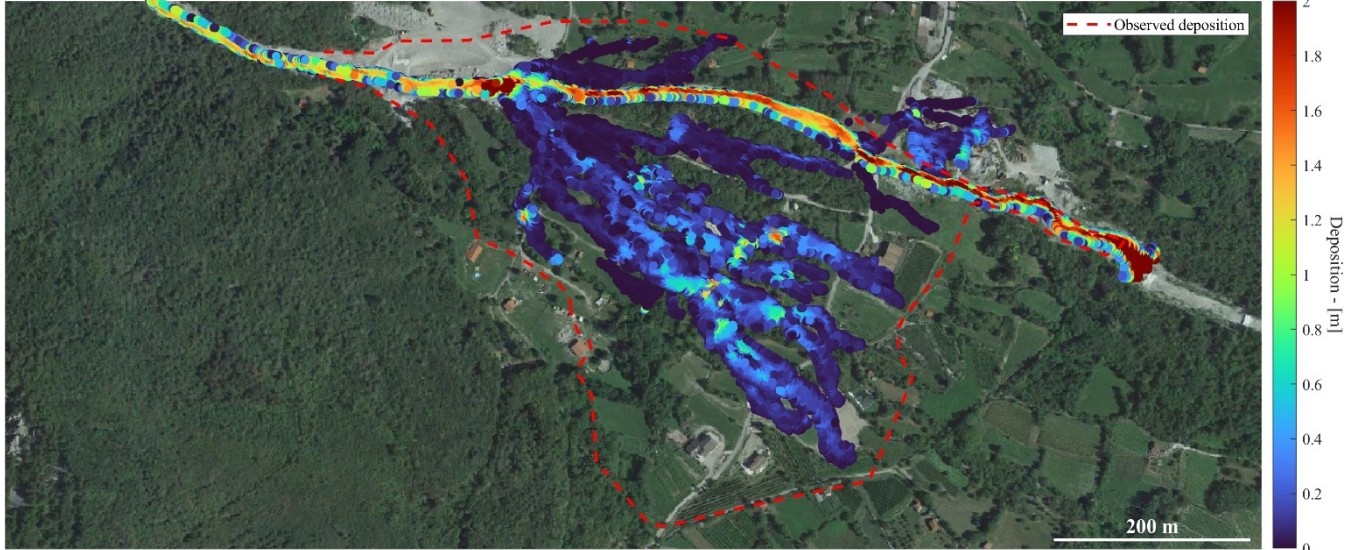

Figure 11. Deposition map at the end of the simulation using the O'Brien model with $C_v = 0.423, \alpha_1 = 0.0036\ Pa\ s, \alpha_2 = 0.001\ Pa, \beta_1 = 20.5, \beta_2 = 25.7, \gamma_m = 26.5\ kN\ m^{-3}, n = 0.1\ s\ m^{-1/3}$ and $K = 2285$ (© Google Earth 2019).

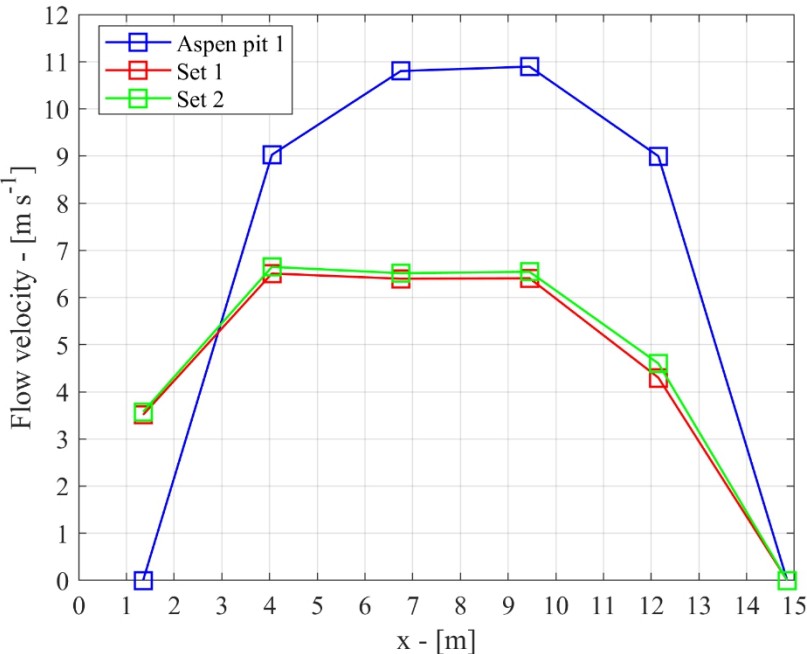

Figure 12. Peak flow velocity module at the measuring cross-section. Aspen pit 1 rheological set is given by $C_v = 0.313, \alpha_1 = 0.0036\ Pa\ s, \alpha_2 = 0.0181\ Pa, \beta_1 = 22.1, \beta_2 = 25.7, \gamma_m = 26.5\ kN\ m^{-3}, n = 0.1\ s\ m^{-1/3}$ and $K = 2285$. Rheological set 1 is given by $C_v = 0.423, \alpha_1 = 0.0036\ Pa\ s, \alpha_2 = 0.0181\ Pa, \beta_1 = 20.5, \beta_2 = 25.7, \gamma_m = 26.5\ kN\ m^{-3}, n = 0.1\ s\ m^{-1/3}$ and $K = 2285$. Rheological set 2 is given by $C_v = 0.423, \alpha_1 = 0.0036\ Pa\ s, \alpha_2 = 0.001\ Pa, \beta_1 = 20.5, \beta_2 = 25.7, \gamma_m = 26.5\ kN\ m^{-3}, n = 0.1\ s\ m^{-1/3}$ and $K = 2285$.

As a second observation, we mentioned above that in the O'Brien's rheology the parameter $K$ always appears multiplied by $\alpha_1$ in Eq. (12) and thus $K$ and $\alpha_1$ are not indipendent and can be considered as a single parameter. This property, that to our knowledge has so far passed unnoticed in the vast literature on FLO-2D and that contributes to limit the calibration effort, is clearly shown by the simple example of Fig. 13, where the 1D profile for a dam break problem on a dry inclined slope for a debris flow described by the O'Brien rheology is shown. The profile is shown $18\ s$ after starting from rest and is computed with two different parameters set. The first is the mentioned Aspen Pit 1 set and the second set is obtained from the first by switching the value of $K$ and $\alpha_1$. As one can observe, there is no detectable difference between the two simulations. It is interesting to observe that using FLO-2D in similar tests, some differences were observed between the simulations, possibly due to other numerical features of that code.

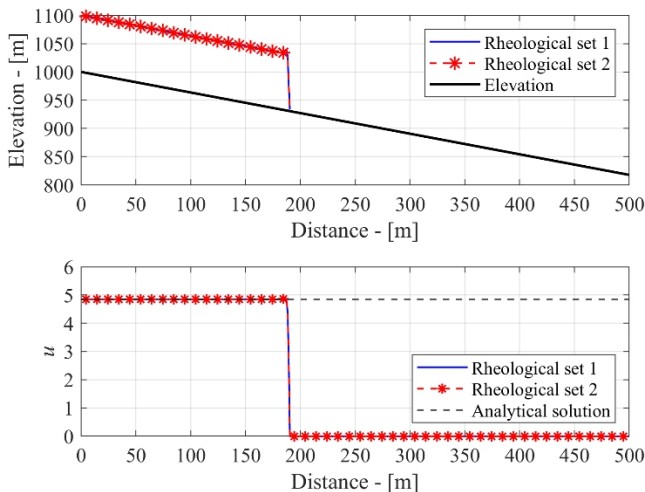

**Figure 13.** Numerical solution of a dam break problem on a sloping dry bed with O'Brien & Julien friction law. The initial condition is a constant fluid depth of $1\ m$ in the first $100\ m$, while completely dry elsewhere. Boundary conditions used are a constant fluid depth upstream and transmissive boundary conditions downstream. The snapshot depicts the numerical solution $18\ s$ after release with two rheological sets: 1) Aspen Pit 1 and 2) Aspen Pit 1 with $\alpha_1 = 2285\ Pa\ s$ and $K = 0.0036$. The dashed black line in the bottom panel depicts the transient analytical solution for the flow velocity showed in Eq. (5). In the top panel the fluid depth has been enlarged for graphical purposes.

## 5 Conclusions

Although monophasic models like FLO-2D, RAMMS, HEC-RAS or TELEMAC-2D only provide a first-order approximation of the complex dynamics of debris flow propagation, they are widely used in practice and depend strongly on parameters that can be difficult to identify. This inevitably leads to a trial-and-error optimization that is computationally demanding. When the number of parameters grows too large, this calibration may become impossible. To limit these "blind" simulations, it is practically very useful to have some simple analytical relations like Eq. (12), (13), (17) and (21) that, in presence of field measurements or other evidence, can be used to evaluate the normal flow velocity for a given parameters set. This velocity is a physical constraint that can be compared to the one expected or observed in the field. The validity on the assumption of a normal flow condition can be locally supported by the comparison with the computed value of the characteristic time. This procedure is shown with reference to the Blé test case, where we tested the two most widely used monophasic models. Using the Voellmy model, one of the parameters, i.e. $\mu$ can be estimated knowing the average slope of the main areas where the debris flow deposited, as suggested in the literature, whereas the parameter $\xi$ can be fixed according to the terminal velocity constraint if such information is available. The O'Brien rheological model does not allow to completely fix each of the parameters on the basis of the information available in the presented test case. To make a comparative test, we fixed both the terminal velocity and the characteristic time to the same values provided by the calibrated Voellmy model, by focusing on the most important rheological parameters suggested by the Sobol's sensitivity analysis. The results of the comparison show that for the Blé event, the Voellmy model predicts a more realistic deposition map with respect to the O'Brien model. For this model, we show how, starting from an ineffective standard parameterization, it is possible to significantly improve the performance of the model. The proposed equations (12), (13), (17) and (21) can also provide additional insights on the physical interplay between parameters, like the previously undocumented lack of observability of the O'Brien rheology with respect to $K$ and $\alpha_1$. These two parameters appear only in a multiplicative fashion within the equations and, accordingly, can be considered as a single parameter ($\alpha_k = \alpha_1 K$), reducing by one the degrees of freedom during the calibration process. The Sobol's sensitivity analysis is performed to highlight which parameters have a greater influence on the uniform velocity for each rheology, so further helping the calibration effort. Regarding the O'Brien rheology, the most influential parameters turn out to be $C_v$ (silt concentration), $\beta_1$ (exponential coefficient of the viscosity) and $n$ (Manning's coefficient). For the Voellmy rheology each parameter is equally important. In the Bingham rheological model the viscosity parameter, i.e. $\eta$, dominates the magnitude of the uniform flow velocity. In both O'Brien and Bingham models, the viscosity that appears in the linear term in

the velocity is very important. Finally, in the case of the Bagnold rheology, the most influential parameter in terms of the uniform velocity is the transport parameter, which traditionally is calibrated experimentally (Armanini et al., 2009). As stated by Zegers et al. (2020), "the development of computationally frugal methods to understand parameter interactions in environmental models emerges as an attractive avenue for future research". Accordingly, our contribution moves in this

direction: we believe that the proposed analytical solutions, along with the sensitivity analysis of the parameters involved, can be a significant help to guide the calibration of these numerical models. While our results are limited to the analysis of a stage-discharge relation and of a characteristic time, so providing a single piece of information of a more complex picture, they are not confined to a particular case or geometry and accordingly have a wider applicability. Other possible uses of the proposed solutions (Eq. 12, 17 and 21) can be in the implementation of boundary conditions for numerical solver of monophasic models,

in the computation of stage-discharge equations at a given cross-section (as showed in the Blé application, i.e. Eq. 24) or even, when coupled with a local mass balance, in the implementation of simplified kinematic routing schemes that could be used to provide back-of-the-envelope evaluations of the debris flow potential along the drainage network of a watershed.

### Code availability

All software used in this research is available upon reasonable request to the authors.

### Data availability

No datasets were used in this article.

### Author contribution

RB and MP conceptualized the work, RB carried out the formal analysis and all authors contributed to write the paper, finally, GF reviewed and improved the paper.

### Competing interests

The authors declare that they have no conflict of interest.

### Acknowledgements

This research was accomplished within the project "Dynamics of debris flows in Val Camonica valley (Brescia): field monitoring of the Val Rabbia and Blé debris flow catchments", funded by Regione Lombardia, Italy. We thank Prof. Berti for

leading the experimental activities in the Blé creek that provided the value of the measured debris-flow discharge used in the paper.

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
