# Peer review of "Use of simple analytical solutions in the calibration of Shallow Water Equations debris flow models"

_EGUsphere, 2024_

## Referee Comment (RC1)

General comments

The Authors proposed analytical/numerical solutions for different rheological models and proposed the Sobol's global sensitivity analysis to evaluate the influence on the simulations of the parameters involved in the investigated models. Their approach can be very helpful to guide the calibration of these numerical models that in the majority of the cases lead to a trial-and-error approach. To improve the paper quality, I would suggest small changes in the paper organization. For example, I'd suggest moving the description of the Sobol analysis in a small, dedicated Section before the models description. Now, the latter is described in the Section of the O'Brien and Julien's model. Furthermore, it would be amazing if the authors would apply O'Brien and Julien's model to simulate the case of Valle Camonica. In this way, coupling the analytical solutions and the Sobol analysis the authors would emphasize their work. Differently, using the 1D case, they should investigate some other parameters with a greater variance than the used one. Generally, I think that Section 4 (Application and discussion) should be strengthened.

Specific comments

Line 31 - 33. Moreover, …has been accomplished. Please, consider deleting or rewriting this sentence. It's not very clear.

Line 43. …we show that two parameters of the widely used FLO-2D. Can you briefly introduce these two parameters?

Line 50. Maybe it should be better to call the flow velocity 'depth-averaged flow velocity'

Line 61.  Probably, the Voellmy's rheology was applied to debris flows before the paper you mentioned (Kelfoun et al., 2011, https://doi.org/10.1029/2010JB007622). Please, check if other authors have already used this approach to model debris flows.

Line 65 – 75. Please, declare θ and Φ in Eq. 4 and 5.

Line 98. Why did you say that the Manning's coefficient can be easily identified? How do you calculate it?

Line 139. Could you insert the Sobol equation?

Line 147 – 148. …obtained considering typical values from the literature. For researchers working on this topic your $C_v$ and $\chi_s$ are reasonable but probably it would be better to motivate their ranges choice.

Line 295. Please, indicate the DEM resolution

Line 295. So, you calculated μ using the slope of the depositional area. However, μ changes during the flow motion. Probably, the procedure more reasonable should be using your analytical solutions to restrict the variability range of μ. Starting from this range, a trial-and-error procedure should be performed to demonstrate that the selected μ (0.249) results in the best match between simulated and real data.

Line 330. 30 s or 18 s?

Technical corrections

Line 76 – 78. Please check the English of the sentence 'The solution for.. when B = 0.

Line 167. Zegers et al. (2020),  which

---

## Author Comment (AC1)

**Reviewer 1**

First of all we would like to thank this reviewer for his/her very supportive evaluation of the paper that demonstrates that the topic is of strong interest for researchers active in the area of hydraulic hazard in mountain regions.

- "For example, I'd suggest moving the description of the Sobol analysis in a small, dedicated Section before the models description. Now, the latter is described in the Section of the O'Brien and Julien's model."

Done: Section 3.1 (Sobol's global sensitivity analysis) is now dedicated to the description of the Sobol indices used in the paper to assess the sensitivity of the rheological models to each parameter.

- "Furthermore, it would be amazing if the authors would apply O'Brien and Julien's model to simulate the case of Valle Camonica. In this way, coupling the analytical solutions and the Sobol analysis the authors would emphasize their work."

Following your suggestions, as well as a similar request from the other reviewer we added additional two sections in the paper with the calibration of the Bingham (section 4.2) and O'Brien (section 4.3) rheology.

- "Differently, using the 1D case, they should investigate some other parameters with a greater variance than the used one. Generally, I think that Section 4 (Application and discussion) should be strengthened."

We strengthened Section 4 as suggested by the Reviewer. Regarding his/her first request, it is not clear which case the reviewer is referring to. We remain available to add modifications in front of a more specific request.

- "Line 31-33. Moreover, … has been accomplished. Please, consider deleting or rewriting this sentence. It's not very clear."

The selected sentence has been deleted as you requested.

- "Line 43. …we show that two parameters of the widely used FLO-2D. Can you briefly introduce these two parameters?"

The two parameters $K$ $[-]$ and $\alpha_1$ $[Pa\ s]$ are now briefly presented in the introduction section.

- "Line 50. Maybe it should be better to call the flow velocity depth-averaged flow velocity"

Done. The definition you highlighted has been changed according to your comment.

- "Line 61. Probably, the Voellmy's rheology was applied to debris flows before the paper you mentioned (Kelfoun et al., 2011, https://doi.org/10.1029/2010JB007622). Please, check if other authors have already used this approach to model debris flows."

We thank the reviewer for this suggestion: now, in addition, we added some other references of authors which used the Voellmy's rheology to model debris flows before the cited one.

- "Line 65-75. Please declare $\vartheta$ and $\varphi$ in Eq. 4 and 5."

Parameters $\vartheta$ and $\varphi$ are now declared explicitly in the text, as you requested.

- "Line 98. Why did you say that the Manning's coefficient can be easily identified? How do you calculate it?"

In the authors' opinion, considering that the Manning's coefficient is a parameter widely used in flood modelling, its value can be estimated according to the tabulated values existing in literature (e.g. for a non-comprehensive list, Chow, 1959; Bray, 1979; Jarret, 1984), and finding the best match between the kind of channel type and description reported and the characteristics of the channel or conoid under investigation. Similarly, many papers are available that link the Manning's coefficient to the soil cover for flow over vegetated surfaces. Accordingly, unlike other parameters like the sediment concentration, the Manning's coefficient is a reasonably fixed value that does not change between various event occurring in the same area.

Bray, D. I.: Estimating average velocity in gravel-bed rivers: American Society of Civil Engineers, Journal of the Hydraulics Division, v. 105, no. HY9, p. 1103-1122, 1979.

Chow, V. T.: Open Channel Hydraulics. McGraw-Hill, New York, 1959.

Jarrett, R. D.: Hydraulics of high-gradient streams: American Society of Civil Engineers, Journal of Hydraulic Engineering, v. 110, no. HY11, p. 1519-1539, 1984.

- "Line 139. Could you insert the Sobol equation?"

Done. Sobol first order and total order indices definitions are now shown in Section 3.1 (Sobol's global sensitivity analysis).

- "Line 147-148. …obtained considering typical values from the literature. For researchers working on this topic your $C_v$ and $\gamma_s$ are reasonable but probably it would be better to motivate their ranges choice."

We thank the reviewer for this observation, now additional references are provided in Table 2

which explain the ranges of the cited parameters.

- "Line 295. Please, indicate the DEM resolution"

The requested information has been included inside section 4 of the paper.

- "So, you calculated μ using the slope of the depositional area. However, μ changes during the flow motion. Probably, the procedure more reasonable should be using your analytical solutions to restrict the variability range of μ. Starting from this range, a trial-and-error procedure should be performed to demonstrate that the selected μ (0.249) results in the best match between simulated and real data."

We thank the reviewer for this comment. Additional simulations of the Ble event have been performed in which the $\mu$ parameter has been modified in order to prove that the selected value ($\mu = 0.249$) is probably the best match between simulated and real data. Fig. 1 shows the boundary of the deposition map obtained by first setting $\mu = 0.19$, which corresponds to a deposition angle of about 10° (slightly less with respect to the average slope of the main deposition areas highlighted in the paper of about 14°). The value of $\xi = 45\ m/s^2$ was then obtained by using Eq. (23) of the revised paper. It is clear from Fig. 1 that the selected parameter set models a debris flow which is too fluid as it flows out of the domain towards areas which have not been affected by the event. On the contrary, Fig. 2 shows the deposition map obtained using $\mu = 0.36$ (deposition angle of 20°, slightly higher with respect to the average slope of the main deposition areas, i.e. 14°) and $\xi = 87\ m/s^2$. The value of $\xi$ has been obtained by imposing a terminal velocity of 5.4 $m/s$ at the measuring station using Eq. (23) of the revised paper, as done for the previous case. Now the debris flow is too viscous as it even fails to reach the main deposition areas observed in the field. Note that even by changing the rheological parameters, the depth-averaged velocity of the debris flow at the measuring station is still around $5 - 6\ m/s$, thus confirming the analytical predictions even for multiple rheological sets. We did not add this in the paper due to space limitations.

[Figure]

**Figure 1. Deposition map at the end of the simulation using the Voellmy model with $\mu = 0.19$ and $\xi = 45\ m/s^2$ (© Google Earth 2019).**

[Figure]

**Figure 2. Deposition map at the end of the simulation using the Voellmy model with $\mu = 0.38$ and $\xi = 87\ m/s^2$ (© Google Earth 2019).**

- "Line 330. 30s or 18 s?"

We thank the reviewer for pointing out this inconsistency, now the correct time, i.e. 18 s, is correctly written both in the paper and in the caption of Fig. 9.

- "Line 76-78. Please check the English of the sentence "The solution for.. when $B = 0$.""

The sentence has been modified according to your suggestions: "The solution for the Voellmy model, investigated by Herganten and Robl (2015) as well as Pudasaini and Krautblatter (2022), implemented in the widely used RAMMS software (Christen et al., 2010), can be obtained using Eq. (5) when $B = 0$."

- "Line 167. Zegers et al. (2020),  which"

The modification you suggested has been included in the paper.

---

## Author Response (AR2)

**Associate editor**

- "I have received the comments on the revised version of your manuscript. On this basis, the decision is to accept with minor corrections. Thank you in advance for resolving the reviewer's comments on this latest manuscript version."

We thank the associate editor for her/his positive decision regarding the manuscript. We have now introduced the minor revisions requested by the reviewer. We look forward hearing from you.

**Reviewer 1**

- "Dear authors, sorry that some of my comments were a bit harsh! Now my feeling is that you did a good job in revising and extending your manuscript and would like to recommend it for publication."

We sincerely thank the reviewer for his insights and suggestions that definitely lead to a more solid paper.

- "In Sect. 2, you state that the discriminant B^2-4AC must be positive for achieving an equilibrium velocity. Owing to the signs, I find it a bit confusing since I tend to interpret it in the way that B^2 must be greater than 4AC. This is true, but A is always negative and C positive (if the slope is steep enough to let movement start). So B^2-4AC is practically always positive as long as B not equal 0 or a not equal 0. So the way you wrote it is formally correct, but not perfectly clear what it means. Maybe you can rephrase it."

We modified the sentence you highlighted to:

"Accordingly, the discriminant $B^2 - 4AC$ must be positive for a uniform velocity to exist, which is always the case provided that all terms are different than zero and $C > 0$, which happens when the slope is steep enough for the motion to occur. Degenerate cases, i.e. when either $A = 0$ or $B = 0$, will be discussed separately."

- "Following the suggestion of the other reviewer, you introduced a separate subsection about Sobol's analysis. This makes sense, but it looks to me as if the following subsection (3.2) about the O'Brien and Julien rheology still contains aspects of Sobol's analysis that apply to the other rheologies, too. Maybe you could streamline this part a bit."

We thank the reviewer for spotting this inconsistency. The mentioned subsection has been streamlined and all general aspects related to the Sobol's analysis have been moved to the corresponding section (3.1 Sobol's global sensitivity analysis) as you can see in the tracked changes version of the work.

**List of relevant changes in the manuscript**

We adjusted the paper according to all the suggestions of the reviewer:

- Lines 67-70 of Section 2 (Governing equations) have been modified according to the reviewer's request.
- Section 3.1 (Sobol's global sensitivity analysis) has been streamlined by including all general aspects related to the Sobol's analysis.